# Modification of Sugar Profile and Ripening in Atemoya (*Annona × atemoya* Mabb.) Fruits through Copper Hydroxide Application

**DOI:** 10.3390/plants12040768

**Published:** 2023-02-08

**Authors:** Caroline P. Cardoso, Felipe G. Campos, Gabriel M. Napoleão, Gustavo R. Barzotto, Lauro P. Campos, Gisela Ferreira, Carmen S. F. Boaro

**Affiliations:** 1Biodiversity and Biostatistics Department, Institute of Biosciences, UNESP—São Paulo State University, Campus Botucatu, Street Prof. Dr. Antonio Celso Wagner Zanin, 250-District de Rubião Junior, Botucatu 18618-689, São Paulo, Brazil; 2School of Agriculture, Plant Production Department, UNESP—São Paulo State University, Campus Botucatu, Ave. Universitária, nº 3780-Altos do Paraíso, Botucatu 18610-034, São Paulo, Brazil

**Keywords:** carbohydrates, custard apple, cupric fungicide, maturation index

## Abstract

Atemoya (*Annona × atemoya* Mabb.), a climacteric fruit of the Annonaceae family, is becoming increasingly popular due to its organoleptic and nutritional properties. Anthracnose, a fungus of the Colletotrichum genus, is one of the most serious diseases in orchards, causing significant damage if not controlled, so producers use phytosanitary products. The current study sought to investigate the quality of atemoya fruits after harvest in an orchard with anthracnose controlled by Cu(OH)^2^ application: T1—no Cu(OH)^2^, T2—7.8 mL Cu(OH)^2^ L1 divided into two applications, T3—15.6 mL Cu(OH)^2^ L1 divided into four, T4—8.0 mL Cu(OH)^2^ L1 divided into eight, and T5—13.0 mL Cu(OH)^2^ L1 divided into thirteen applications. The sugar profile of fruits was examined, as well as MDA, H_2_O_2_, and quality parameters such as pH, mass, soluble solids, titratable acidity, and maturation index. MDA, such as H_2_O_2_, can function as a signal molecule. Eight applications of 1.0 mL L-1 Cu(OH)^2^ resulted in increased concentrations of H_2_O_2_ and MDA, signal molecules involved in sugar modification profiles such as glucose, fructose, and trehalose. It also had a high titratable acidity, a lower maturation index, better fruit quality, and a longer shelf life.

## 1. Introduction

Atemoya (*Annona × atemoya* Mabb., also known as *Annona cherimola* Mill *× Annona squamosa* L.) or custard apple are common names for the pollen donor species sugar apple fruit (*A. squamosa* L.) and pollen recipient species cherimoya (*A. cherimola* Mill.) [1]. Consumption of atemoya (*Annona × atemoya* Mabb.) in Brazil has increased since 2012 [2] due to its widespread use, nutritional properties, health benefits [3,4,5,6,7] and adaptability capacity to consumer demands such as fewer seeds, larger fruit size, and sweetness when compared to sugar apple fruit and cherimoya [1,2,8].

Atemoya was discovered in 1908 by a researcher in the United States, but Florida-grown cultivars originated in Israel [1,8]. The global consumption and interest in are increasing [2,9,10]. The Thompson cv. of atemoya is the most widely produced in Brazil [11], followed by Gefner, Pink’s Mammoth and African Pride [12].

Atemoya is a climacteric fruit that causes an increase in ethylene synthesis as well as a burst in respiratory activity [1,13,14,15]. Respiration is a well-known physiological process that employs sugars as a substrate [14], and it is separated into three major pathways: glycolysis, the mitochondrial tricarboxylic acid (TCA) cycle, and mitochondrial electron transport [16]. Furthermore, copper can alter glycolysis because the ion can modulate invertase [17,18], stimulate hexokinase activation, and inhibit pyruvate kinase, resulting in a carbon flux reprogramming [19]. Copper is also essential for the ethylene-binding activity of ETR1 (ethylene response 1), a receptor protein [20,21], which is a hormone that causes fruit ripening in climacteric fruits. Copper is also detected in amine oxidases during fruit ripening, where it inhibits putrescine (a competitor of ethylene precursor [16]) and increases the level of abscisic acid (ABA) and H_2_O_2_ [22]. ABA functions as a ripening promoter [23]. 

Sugar buildup is one of the major biochemical activities that occur during the fruit maturation phase [23,24], due to glycolysis, which hydrolyzes sucrose into mono and disaccharides with the help of invertases and/or sucrose synthase (sucrose cleavage enzyme) [14,25]. Sugar content varies with plant species [26,27]. Mannose, a monosaccharide, can alter the development of the enzymatic antioxidant defense system, and fructose and glucose (also monosaccharides) can act in plant mechanism defense [28,29]. Similarly, disaccharides such as trehalose and sucrose play a crucial role in stress resistance because both sugars can maintain proteins and lipid bilayers [30]. Trehalose, on the other hand, can preserve fruit quality by modulating the metabolism of soluble sugars, releasing two glucose molecules to meet energy demands [31,32], and it is 45% sweeter than sucrose [30]. 

As a result, understanding fruit ripening is critical, because this increase in respiration and ethylene synthesis triggers multiple pathways, resulting in changes in fruit quality that include organoleptic and nutritional aspects [33,34,35]. These factors include the category known as compositional in the shelf life of fruits [36], which is particularly important because the commercial value of atemoya fruits is based on quality features [37,38,39]. According to these organizations, an atemoya fruit lot can be divided into three classes: “extra” (virtually no fault, elevated bulk, good uniformity), “I” and “II” with more tolerance conditions and lower value. 

Anthracnose is a disease that is widely seen in atemoya orchards. It is caused by a pathogen of the *Colletotrichum* genus and affects the plant’s development and productivity, such as causing fruit depreciation by the formation of dark spots on its surface [10,40]. This disease is more common in areas with high relative humidity, and it causes 53 to 70% of fruit loss when there are lengthy periods of rain during blooming and fruit formation [40,41]. In conclusion, parasite infection is stressful and generates metabolic imbalance in plants [42,43].

Farmers tend to employ phytosanitary products as control management to ensure disease-free fruit production with high standards and higher value. There are currently eleven commercial products (fungicides) licensed in Brazil for Annonaceae, one of which is copper hydroxide, a well-known active component [44]. 

Copper’s range of activity includes various phytopathogenic microorganisms, according to La Torre et al. [45], and these microorganisms do not exhibit copper resistance, as reported by the Fungicide Resistance Action Committee (FRAC) [46].

Copper fungicides, such as Cu(OH)_2_, have a low solubility and are absorbed on the surface of leaves, where it combines with secreted exudates such as malate, allowing their absorption by fungi and plants [45,47]. The pathogen is directly affected by this molecule due to the pathogen’s minimal demand for copper for physiological processes, which, when in excess, inhibits enzymes required for its survival [48].

Copper occurs as Cu^2+^ and Cu^+^ under physiological conditions [49], and in the presence of water or any other abiotic factor that facilitates its contact with microorganisms, copper enters the cytoplasm owing to dissolution and it may prevent disease [45]. The ions block enzymes not dependent on the sulfhydryl group, such as catalase, sucrase, arginase and betaglucosidase, causing a general metabolic problem and cellular integrity disruption [50,51]. 

Because copper stimulates lignin biosynthesis in plants [43], its deposition in the cell wall serves as a physical barrier against pathogen infection and is thought to be a protective mechanism [52]. Previous research has shown that lignin can scavenge free radicals, stabilizing reactive oxygen species (ROS)-induced processes [53,54]. As a result, copper can induce salicylic acid [19,55], a hormone that is also induced by ROS [56]. 

Given the foregoing, the current study sought to assess the postharvest quality of atemoya cv. Thompson fruits in an orchard with anthracnose controlled by Cu(OH)_2_ spraying. 

## 2. Results and Discussion

### 2.1. Hydrogen Peroxide and Lipid Peroxidation in Fruits

At 175 days after the first Cu(OH)_2_ application (DAFA), atemoya fruits sprayed with different concentrations of Cu(OH)_2_ showed higher concentrations of hydrogen peroxide in comparison to T1 (control) (Figure 1a), possibly indicating a high activity of copper amine oxidases during ripening, as found in previous studies [22]. Furthermore, copper as Cu^2+^ can inhibit catalase, an enzyme with the ability to directly break H_2_O_2_ into H_2_O and O_2_, resulting in high H_2_O_2_ concentrations in plant tissue [57,58,59], a trend that was also observed in this investigation (Figure 1a).

Because all plants treated with Cu(OH)_2_ had it regulated, the copper application in fruits in the current investigation may have caused H_2_O_2_ synthesis and may have aided to denature the fungal cell wall [60,61]. Furthermore, hydrogen peroxide can alter *C. gloeosporioides*’ infection approach, requiring the fungus to employ a subcuticular, intramural necrotrophic strategy [62]. 

Plants respond with an oxidative burst appearance depending on copper content, as demonstrated in *Citrus aurantium* L. plants treated with the ion [63]. The current study’s oxidative burst could have been caused by a combination of pathogen attack and copper administration. Because of the ions’ capacity, copper treatments produced significant H_2_O_2_ concentrations. H_2_O_2_ can be damaging to plants and fruits in excess [64], but at 175 DAFA, the fruits were not visibly affected, and it could have acted as a signal molecule. 

Treatments 2 and 4 produced fruits with more lipid peroxidation and higher malondialdehyde (MDA) buildup than the other treatments (Figure 1b). Due to lipoxygenase activity as a result of pathogen incidence [65] and a high amount of H_2_O_2_, a ROS, these two treatments may have produced lipid peroxidation twice (Figure 1a). This scenario is reliable since both treatments received less copper when compared to T3 and T5. T1 (control) remained low since its H_2_O_2_ concentration in fruits was already the lowest of all, even when its MDA content was compared to T3 and T5 fruits. The similar quantity of MDA can be explained because, in some situations, this aldehyde may operate as a protective mechanism rather than a signal of damage [65,66]. Even though the MDA concentration in T2 and T4 fruits was higher than in the others, it did not induce eye damage. MDA could also have served as a quality indicator in fruit.

MDA content in the spinach variety Japanese big leaf increased with 800 mg L^−1^ of CuSO_4_, but the mentioned concentration and higher did not terminate plant development [67]. Thus, MDA concentration did not differ in carrots given a maximum of 400 mg kg^−1^ of pure Copper(II) nitrate trihydrate (Cu(NO_3_)_2_ 3H_2_O) [68]. These remarks, along with this work utilizing lower copper concentrations, clearly suggest that MDA, such as H_2_O_2_, can operate as a signal molecule. 

### 2.2. Carbohydrates

Different quantities of copper caused a significant concentration of hydrogen peroxide (Figure 1a) which may have contributed to the divergence of sugar profiles (Figure 2). This sugar profile alteration was expected because copper participates in the respiration process as a signaling cascade inducer, cofactor of enzymes, and polyamine cofactor [19,22,69], and plants were diagnosed with anthracnose. As a result of the high quantity of hydrogen peroxide, carbohydrates may have accumulated and served as signaling molecules.

T4 atemoya fruits had higher concentrations of fructose, glucose, and trehalose than T1 (control) (Figure 2a,b,e). Because copper supply can reduce invertase activity, elevated fructose and glucose concentrations may reflect high activity of sucrose synthase rather than invertase [70]. According to a previous review [71], high concentrations of both monosaccharides (fructose and glucose) in T4 fruits indicate that sugars can induce callose deposition, which acts as a physical barrier through cell wall reinforcement due to pathogen attack. Furthermore, the combination of glucose with copper increases antioxidant activity [72], which aids in ROS scavenging over time. Fructose is also sweeter than glucose and sucrose [73]; therefore, the high carbohydrate concentration in fruits adds value.

Copper hydroxide application in T4 may have contributed to a rise in trehalose in fruits, improving their quality and acting as an elicitor of defense systems [74]. Trehalose may be appealing to consumers because it is 45% sweeter than sucrose [30]. This organoleptic element may pique the interest of customers. Furthermore, the high trehalose content owing to copper treatment provided a chance to study changing climatization because trehalose, such as sucrose, serves as a cryoprotectant [30], and atemoya’s climatization is complex. 

T5 fruits had the greatest sugar concentration of any fruit (Figure 2d). Sucrose is the primary carbohydrate transported from the plant to the fruits [75], and it plays an important role in the preservation of the quality of fleshy fruits [33], as hexoses are required to generate energy and act as a signal molecule, inducing or inhibiting the synthesis of sucrose, cellulose, proteins, glucose, and antioxidant compounds [76]. Copper caused a change in carbon flux [19], and it is possible that repeated copper treatments inhibited invertase and sucrose synthase in atemoya fruits, resulting in high sucrose concentrations. 

When carbohydrates are continuously provided, chlorophyll degradation can be re-strained [29,77,78], resulting in a longer shelf life due to a thicker cell wall (skin stiffness) and antioxidant activity [25,34]. As a result, copper is thought to have aided plant defense mechanisms by activating plant-specific genes [19,22]. Sugar in fruits can enhance soluble solids, skin stiffness, catalase, and decrease the action of glucose phosphate isomerase and cytochrome oxidase [18,34,35], which could explain why fruits have a longer shelf life (Figure 3e). Because cytochrome c oxidase requires copper for catalysis, biogenesis, assembly, and stability [79], copper application may have impacted this mitochondrial complex.

### 2.3. Fruit Quality

All treatments produced fruits weighing more than 300 g at 175 and 178 DAFA (Figure 3a), the time when fruits are commercialized in large cities. The findings confirm that all the fruits in this study could be marketed, meeting the minimum mass value specified by Brazilian, Asian, and European regulations [37,38,39]. Thus, our findings are consistent with previous research on atemoya fruits free of anthracnose [80,81,82]. T1 (control) fruits, on the other hand, were the first to develop petiole fissures as a result of atemoya ripening [8,11]. This flaw may jeopardize the fruits’ ingestion.

T3 fruit mass was found to be greater than other treatments, with masses greater than 400 g until 184 DAFA (Figure 3a), although it did not show enhanced sugar concentrations except for mannose (Figure 2c). It is suggested that 15.6 mL L^−1^ of Cu(OH)_2_ divided across four administrations aided in the maintenance of fruit mass.

T5 fruits, on the other hand, had a low mass, confirming that copper influences respiration increase. When compared to T3, this increase resulted in smaller fruits; the sucrose pathway appeared to be reconstructed in the different treatments, proportioning fruits organoleptic alterations and component concentrations.

Soluble solids include water-soluble substances such as carbohydrates, vitamins, acids, and amino acids, and their concentration tends to rise due to polysaccharide breakdown [83]. In general, soluble solids increased up to 181 DAFA (Figure 3b), most likely due to an increase in soluble sugars from the transformation of starch into sugars during the maturation phase [83]. Interestingly, recent research on strawberries (*Fragaria ananassa* Duch.), a non-climacteric fruit, linked an increase in soluble solids to a decrease in fruit water potential during fruit ripening, indicating that a decrease in fruit water potential would be a primary signal contributing to fruit ripening onset [84]. 

The data in the current investigation showed no difference between treatments, except at 184 DAFA when T3 had lower soluble solids than T1 (control) and T5 (Figure 3b); however, several fruits of T1 had fluid leaks and cracks from the petiole, as previously mentioned, and consumer avoidance would be expected. Thus, at 184 DAFA, T3 fruits had the highest mass value (Figure 3a), indicating that plants could have reprogrammed carbon flow with copper aid [19]. The decrease in soluble solids at 184 DAFA could indicate that simple sugars are being used in cellular respiration.

Copper hydroxide applications to control anthracnose did not alter the soluble solids in general, indicating that its use is safe for fruits. However, the sugar profile changed across all treatments (Figure 2).

When compared to other treatments, fruits from T5 at 175 DAFA exhibited an acid pH (Figure 3c). Because items with higher acidity are naturally more resistant to deterioration [85], it has been proposed that applying 13 mL Cu(OH)_2_ L^−1^ split thirteen times (T5) increased the acidity of the fruits, resulting in a longer shelf life. Because the harvest occurs at an early stage and the analysis is not too sophisticated, pH measurement is a crucial parameter to ensure great quality in atemoya fruits. 

Organic acid determination in fruits is an important criterion because of its influence on organoleptic qualities and fruit stability for eating, varied between 0.2 and 0.3% in low acidity fruits [85], as in the case of atemoya species. In general, the titratable acidity of fruits treated with Cu(OH)_2_ was higher in this investigation than in previous studies, whereas fruits in T1 (control) showed average values [81,86,87]. 

The average value in soursop (*Annona muricata* L.) fruits is elevated, and its titratable acidity can reach 0.72% [88], although the T4 and T5 results in the current study remain increased (Figure 3d). When copper levels above the ideal range for plant species, it can interfere with respiration, resulting in high concentrations of organic acids and dominant carbon fluxes throughout the ripening phase [69]. Also, copper nanoparticles in fruits reduce titratable acidity, extending shelf life and stimulating bioactive component accumulation in tomato (*Solanum lycopersicum* L.) fruits [89].

T4 and T5 fruits had higher titratable acidity (Figure 3d) and a lower maturation index (Figure 3e), implying a longer shelf life than other treatments. The inverse relationship between titratable acidity and fruit maturity index during abiotic stress is well recognized [90]. As a result, lower Cu(OH)_2_ concentrations and a high application frequency (T4 and T5 cases) reduced the maturity index of atemoya fruits, delaying maturation and extending shelf life. 

According to Batten [91], peel browning evaluation revealed fruits with a grade of three (Figure 3f), indicating fruits acceptable for marketing. Notes equal to or more than three were observed in all treatments until 181 DAFA (Figure 3f). Copper hydroxide in 8.0 mL L^−1^ divided into eight applications (T4) revealed fruits that had less than 25% of their peel browned until 184 DAFA. T4 had darkened skins with less than 25% of their surface at 181 DAFA, when atemoya fruits showed their maximum maturation rate (Figure 3e). A homogenous lot of T4 fruits in terms of peel browning could be obtained, which is an important feature for fruit dealers due to reduced handling. 

It is crucial to note that having homogeneous fruits for a longer period lowers the need for consumers to choose and avoids unnecessary commercialization handling; therefore, homogeneity is a desired attribute in many fruits since it helps to retain fruit quality [92,93]. Despite the norm established by Batten [91], customer demand is high, and it necessitates fruits with no or few faults. As a result, suitable fruits may be wasted, and consumer reeducation is recommended, because fruits can still provide suitable flavor and nutritional characteristics despite their visual appearance. 

High quantities of sugars (trehalose, sucrose, and glucose), organic acids, MDA, and H_2_O_2_ may alter fruit quality [19,35,65]. 

### 2.4. Heat Map

Three clusters were discovered using hierarchical cluster analysis (HCA) (Figure 4). Cluster I has the bulk of T1 (control) fruits, Cluster II contains all T4 fruits, and Cluster III contains T3 fruits. T2 and T5 fruits were strewn about in these three bunches. 

T5 fruits had a negative connection between sucrose concentration and pH, implying that pH may have induced sucrose or vice versa (Figure 4). Sucrose promotes invertases, which operate as hydrolyzing enzymes and have transferase activity [17] and this negative relationship between pH and sucrose level has already been documented [94]. As a result, immature fruits contain more acid invertases than older fruits. There was also a negative relationship between pH and soluble solids and T5 fruits, which may have been suppressed by the micronutrient due to copper’s modest but frequent supply [70]. 

Glucose, fructose, trehalose, and MDA all showed positive relationships in T4 fruits, indicating that MDA can operate as a signal molecule during the early stages of fruit ripening by activating regulatory genes involved in plant development and defense [65]. In contrast, H_2_O_2_ had a negative connection with mannose, sucrose, and T4 fruit mass. Because hydrogen peroxide can boost invertase activities depending on concentration [95], increasing fructose and glucose content, these data could indicate an influence of this ROS content on the amount of carbohydrates. 

## 3. Materials and Methods

The experiment was conducted in a commercial orchard of atemoya plants (*Annona × atemoya* Mabb.) cv. Thompson with anthracnose symptoms, in Botucatu municipality, in the state of So Paulo, located 22°59′31″ S and 48°28′28″ W, in a property entitled Estância Rio Sul.

The Köppen–Geiger categorization of the region’s climate is mesothermal humid Cfa, with an average monthly temperature over 22 °C and an annual rainfall of 1377 mm [96]. 

The experimental area was established using a commercial orchard (17 years old) of atemoya cv. Thompson with 30 plants spaced 4.1 m between plants and 3.6 m between rows. Fertilizations were carried out prior to the experiment based on the findings of soil chemical analyses, in accordance with the nutritional needs of temperate climate plants II [97] and “anona” chapter on “Boletim 200” [98]. The same technical bulletins were followed by cultural practices. 

Between 22 November 2020, and 31 May 2021, the experiment was carried out. It used a randomized block design with four replications and one plant per plot. The study evaluated biweekly foliar applications of copper hydroxide and/or water + adjuvant, in different concentrations and application numbers, based on the recommendation of a product registered by the Ministry of Agriculture, Livestock, and Supply (known as MAPA due to Portuguese initials) in Brazil [44] for the control of anthracnose in Annonaceae (Table 1).

A backpack sprayer (TRATO^®^, São Paulo, SP, Brazil, TP20) was used to apply fungicides throughout the canopy of all plants, using 1.5 L of a mixture of copper hydroxide and Haiten^®^ (0.01 mL L^−1^ of water) at neutral pH. Applications began on November 22nd, 2020, in plants with late flowers and fruits in the early stages and terminated on May 9th, 2021, 168 days after the initial application of Cu(OH)_2_ (DAFA).

At 175 DAFA, and with only seven days to go, five atemoya fruits were collected at the harvest point of each repetition, for a total of 100 fruits. All assessments were carried out at the Biodiversity and Biostatistics Department of the UNESP Biosciences Institute, Campus Botucatu, São Paulo, Brazil (22°49′10″ S, 48°24′35″ W).

Despite the lack of data, anthracnose incidence assessments were carried out using the methods suggested for mango (*Mangifera indica* L.) [99], because there is no guidance in the literature for atemoya. Two evaluations were performed, the first before the start of Cu(OH)_2_ (0 DAFA) applications and the second after the end of all applications (225 DAFA), once the experiment was extended to gas exchange measures in atemoya plants. For each evaluation period, three plants were sampled from each treatment. When different Cu(OH)_2_ concentrations were administered to plants that had anthracnose in the initial evaluation (0 DAFA), the disease was controlled (not yet published). 

According to Alexieva et al. [100], the H_2_O_2_ content was evaluated using trichloroacetic acid extraction and measured at 390 nm. Thio-barbituric acid (TBA) and trichloroacetic acid were used to perform lipid peroxidation, which was measured at 560 and 600 nm [101]. 

Carbohydrates were measured in immature fruits immediately after harvest in the current investigation. Total soluble sugars were extracted from 100 mg of macerated fruits using 80% ethanol and supernatants from three extractions [102]. For total quantification, the soluble sugar profile was evaluated using high-performance ion chromatography (Dionex ICS−5000^+^) for total quantification. All 20 soluble sugar samples and standards were filtered via a 0.22 m filter before being examined in a chromatography system outfitted with a quaternary pump, automated samples, an electrochemical detector DCS5000 (Thermo^®^, Waltham, MA, USA), column P100 (Carbopack^®^, Valinhos, SP, BR), gold and Ag/AgCl reference electrodes. For 35 min, the eluent phases A, B, and C were 640 mM sodium hydroxide, 0.5M sodium acetate, and ultrapure water, with a flow rate of 0.7 mL min^−1^. The injections of the samples were not repeated.

The identification of soluble sugars was accomplished by comparing the peak retention times of all 20 samples to standard retention times (trehalose, mannose, glucose, fructose, and sucrose; Sigma-Aldrich HPLC grade, 99.9% purity) and co-injecting a standard solution with the sample. A calibration curve was used to calculate carbohydrate concentration (Appendix A) in mg g^−1^ fresh fruit mass.

Fruits were stored at room temperature for 15 days after harvesting, with destructive evaluations performed every three days. The following criteria impacted the quality of the fruits. 

Fresh fruit mass (g) measured using a precise scale (0.0001 g). According to AOAC [103], soluble solids (°Brix) were assessed using a digital refractometer and three drops of the juice’s center pulp. Titratable acidity was evaluated using 5 g of fruit extract diluted in 50 mL of deionized water, titrated with a standard solution of 0.1 M NaOH, and phenolphthalein as an indicator, according to AOAC [103]. The results were given in g citric acid per 100 g fresh mass^−1^. The maturation index was calculated using the ratio of soluble solids to titratable acidity. The hydrogenic potential (pH) of fruits was evaluated using a pH meter and the AOAC procedure [104]. The darkening of the fruit skin was assessed using Batten’s scale for *Annona × atemoya* Mabb [91]. 

Heat maps were created using Heatmapper software [105], making it easier to observe variable and treatment relationships.

Data were subjected to Levene’s test to ensure variance homogeneity and the Kolmogorov–Smirnov normality test. 

At 175 DAFA, hydrogen peroxide, lipid peroxidation, and carbohydrates were all measured. Fruit quality was assessed using a 5 × 5 factorial design, five foliar fungicide application regimens, and five periods after fruit harvest. The F test was used to analyze the variance of the findings, and when there was significance (*p* < 0.05), the means were compared using the Tukey test and the MiniTab^®^ statistical tool. Appendix A contains F-values and Tukey test significance.

## 4. Conclusions

Eight applications of 1.0 mL L^−1^ Cu(OH)_2_ resulted in increased concentrations of H_2_O_2_ and MDA, signal molecules implicated in sugar modification profiles such as glucose, fructose, and trehalose. It also had a high titratable acidity, a lower maturity index, greater fruit quality, and a longer shelf life.

## Figures and Tables

**Figure 1 plants-12-00768-f001:**
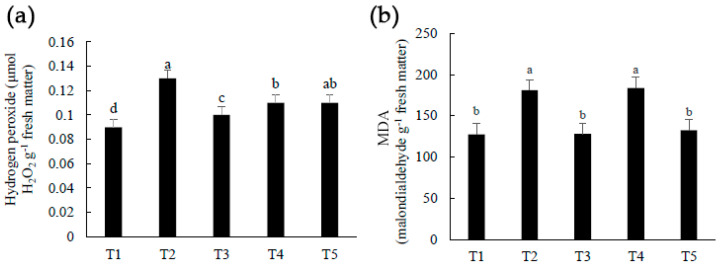
(**a**) Hydrogen peroxide (H_2_O_2_), µmol g^−1^ fresh matter; (**b**) Malondialdehyde (MDA), MDA g^−1^ fresh matter, in atemoya (*Annona × atemoya* Mabb.) fruits cv. Thompson, at 175 days after the first Cu(OH)_2_ application. Medium values. Botucatu, São Paulo, Brazil, 2021. DAFA: Days after the first Cu(OH)_2_ application (plants with flowers and fruits in initial stage). T1 (control)—absence of Cu(OH)_2_; T2—7.8 mL Cu(OH)_2_ L^−1^ of water divided into two applications; T3—15.6 mL Cu(OH)_2_ L^−1^ of water divided into four applications (the recommended dose by MAPA); T4—8.0 mL Cu(OH)_2_ L^−1^ of water divided into eight applications; and T5—13.0 mL Cu(OH)_2_ L^−1^ of water divided into thirteen applications. Means followed by the same letter do not differ from each other by the Tukey test at 1% probability.

**Figure 2 plants-12-00768-f002:**
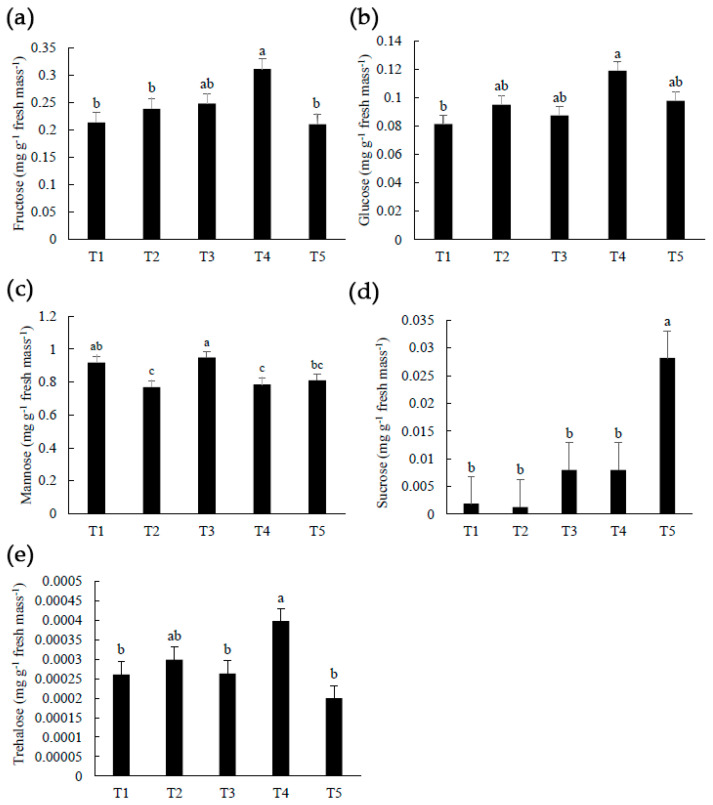
(**a**) Fructose, mg g^−1^ fresh matter; (**b**) Glucose, mg g^−1^ fresh matter; (**c**) Mannose, mg g^−1^ fresh matter; (**d**) Sucrose, mg g^−1^ fresh matter; (**e**) Trehalose, mg g^−1^ fresh matter, in atemoya (*Annona × atemoya* Mabb.) fruits cv. Thompson, at 175 days after the first Cu(OH)_2_ application. Medium values. Botucatu, São Paulo, Brazil, 2021. DAFA: Days after the first Cu(OH)_2_ application (plants with flowers and fruits in initial stage). T1 (control)—absence of Cu(OH)_2_, T2—7.8 mL Cu(OH)_2_ L^−1^ of water divided into two applications; T3—15.6 mL Cu(OH)_2_ L^−1^ of water divided into four applications (the recommended dose by MAPA); T4—8.0 mL Cu(OH)_2_ L^−1^ of water divided into eight applications; and T5—13.0 mL Cu(OH)_2_ L^−1^ of water divided into thirteen applications. Means followed by the same letter do not differ from each other by the Tukey test at 1 and 5% probability.

**Figure 3 plants-12-00768-f003:**
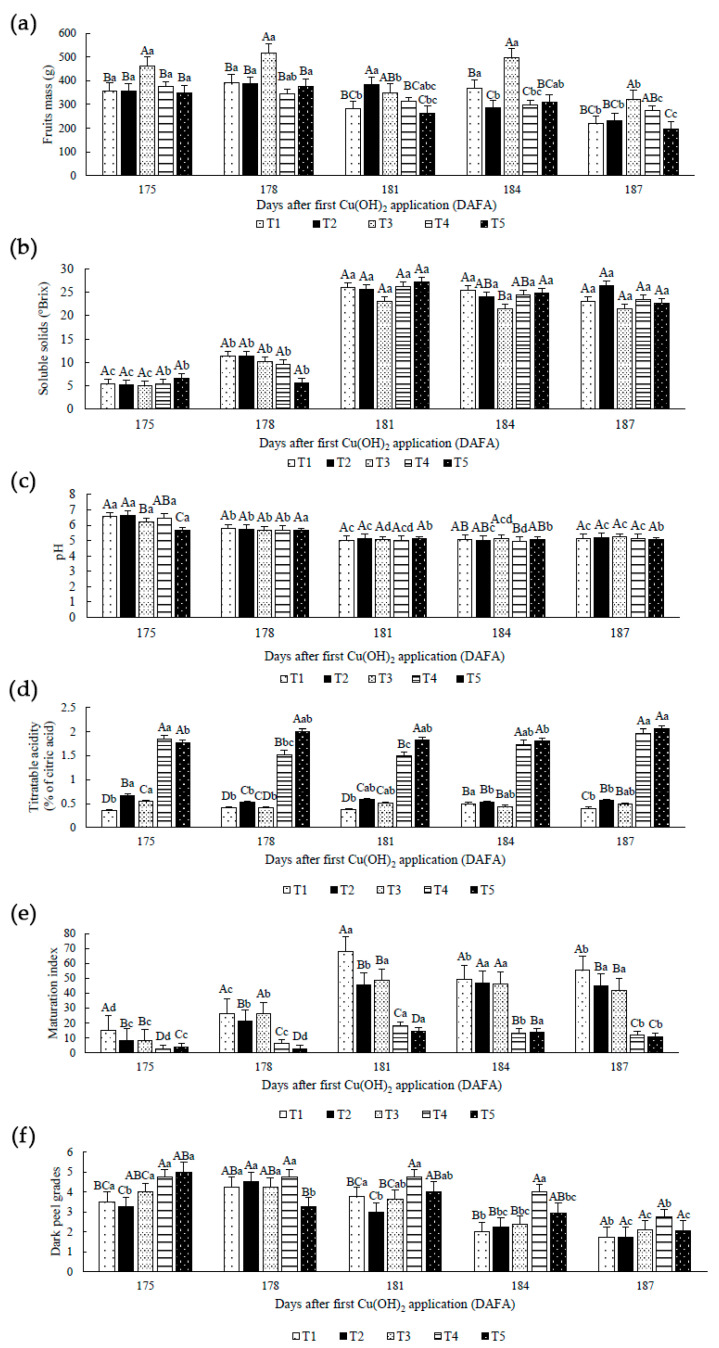
(**a**) Fruits mass (g), (**b**) Soluble solids (°Brix), (**c**) pH, (**d**) Titratable acidity, % of citric acid, (**e**) Maturation index, (**f**) Dark peel grades in 5 evaluations after the first Cu(OH)_2_ application in atemoya (*Annona × atemoya* Mabb.) fruits cv. Thompson. Medium values. Botucatu, São Paulo, Brazil, 2021. DAFA: Days after the first Cu(OH)_2_ application (plants with flowers and fruits in initial stage). T1 (control)—absence of Cu(OH)_2_, T2—7.8 mL Cu(OH)_2_ L^−1^ of water divided into two applications; T3—15.6 mL Cu(OH)_2_ L^−1^ of water divided into four applications (the recommended dose by MAPA); T4—8.0 mL Cu(OH)_2_ L^−1^ of water divided into eight applications; and T5—13.0 mL Cu(OH)_2_ L^−1^ of water divided into thirteen applications. Capital letters represent test treatments in the same DAFA, lowercase letters represent test treatments through time. Bars correspond to average (n = 5). Means followed by the same letter do not differ from each other by the Tukey test at 1 and 5% probability.

**Figure 4 plants-12-00768-f004:**
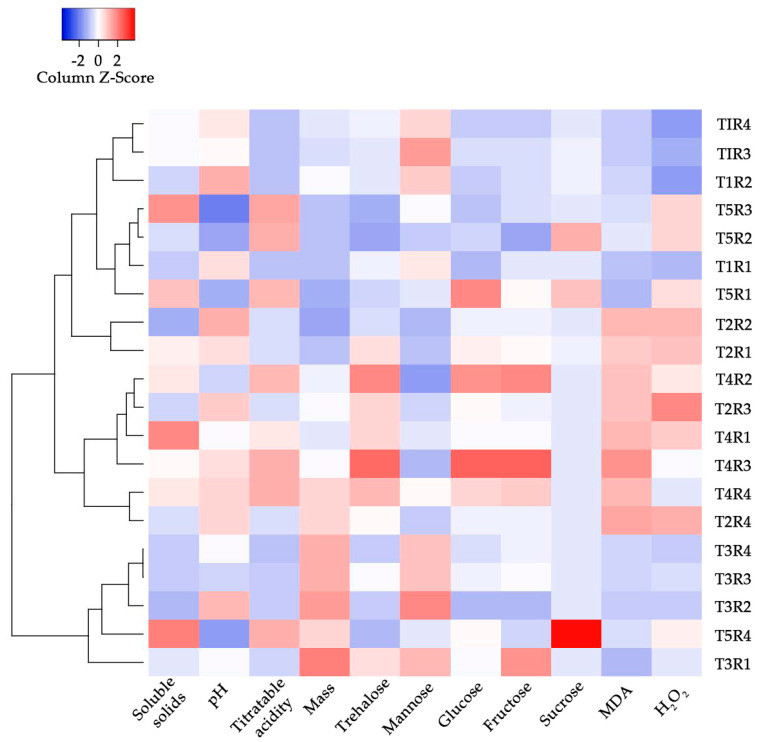
Heat map at 175 DAFA in atemoya (*Annona × atemoya* Mabb.) fruits cv. Thompson. Medium values. Botucatu, São Paulo, Brazil, 2021. DAFA: Days after the first Cu(OH)_2_ application (plants with flowers and fruits in initial stage). T1 (control)—absence of Cu(OH)_2_; T2—7.8 mL Cu(OH)_2_ L^−1^ of water divided into two applications; T3—15.6 mL Cu(OH)_2_ L^−1^ of water divided into four applications (the recommended dose by MAPA); T4—8.0 mL Cu(OH)_2_ L^−1^ of water divided into eight applications; and T5—13.0 mL Cu(OH)_2_ L^−1^ of water divided into thirteen applications.

**Table 1 plants-12-00768-t001:** Discriminated treatments in fruits of atemoya (*Annona × atemoya* Mabb.) cv. Thompson. Botucatu, São Paulo, Brazil, 2021.

Treatment *	Cu(OH)_2_ Concentration per Application (mL L^−1^)	Cu(OH)_2_ Total Concentration (mL L^−1^)	Number of Cu(OH)_2_ Application
T1	0.0	0.0	0
T2	3.9	7.8	2
T3	3.9	15.6	4
T4	1.0	8.0	8
T5	1.0	13.0	13

* T1 (control)—absence of Cu(OH)_2_; T2—7.8 mL Cu(OH)_2_ L^−1^ of water divided into two applications; T3—15.6 mL Cu(OH)_2_ L^−1^ of water divided into four applications (recommended dose by MAPA); T4—8.0 mL Cu(OH)_2_ L^−1^ of water divided into eight applications; and T5—13.0 mL Cu(OH)_2_ L^−1^ of water divided into thirteen applications. Whenever Cu(OH)_2_ wasn’t applied, the same amount of water and adjuvant were applied in the orchard, totaling 13 applications.

## Data Availability

Not applicable.

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
