# Peer review of "Modification of Sugar Profile and Ripening in Atemoya (Annona × atemoya Mabb.) Fruits through Copper Hydroxide Application"

_plants, 2023, doi:10.3390/plants12040768_

Round 1

Reviewer 1 Report

The target crop, atemoya, and the treatment copper, were good research subjects. However, in the introduction, too much unnecessary information was explained in the thesis, and as a result, too many references were cited. In the main text, more than three citations for one fact are indicated, but it is also recommended to reduce these to the most important literature.

Please consider revising the title to ripening because the study did not analyze the determinants that explained shelf-life.

Author Response

Dear,

Reviewer

Plants Journal

We appreciate the opportunity to revise and improve the manuscript following the reviewers’ suggestions. We are grateful for the opportunity to resubmit the manuscript entitled "Modification of sugar profile and ripening in atemoya (Annona × atemoya Mabb.) fruits through copper hydroxide application", n° Submission plants-2156385.

In this new letter we have tried to include all your suggestions and corrections, in order to improve the quality of our study. All reviewers’ suggestions were accepted, and all changes made are highlighted in yellow.

Below you will find the reviewer comments with our responses for each point written in blue color.

Sincerely yours.       

Caroline P. Cardoso, Felipe G. Campos, Gabriel M. Napoleão, Gustavo R. Barzotto, Lauro P. Campos, Gisela Ferreira and Carmen S. F. Boaro.

Reviewer 1

The target crop, atemoya, and the treatment copper, were good research subjects. However, in the introduction, too much unnecessary information was explained in the thesis, and as a result, too many references were cited.

As suggested, we reframed the introduction and modified lines 102-104, as well as the abstract lines 18-27.

In the main text, more than three citations for one fact are indicated, but it is also recommended to reduce these to the most important literature.

As suggested, we reduced the citations and let the most important ones.

Please consider revising the title to ripening because the study did not analyze the determinants that explained shelf-life.

As suggested, we reframed the title. The term ripening from a physiological point of view is more appropriate indeed.

In addition, some parts were ordered in different manner to a better understanding: lines 236-237, 262-273, 277-278, 286-287, lines 333-339 referring all treatments in a table and lines 401-404 reframing the conclusion.

Reviewer 2 Report

Title : need to reframed Modification of sugar profile and shelf life in atemoya (Annona × atemoya Mabb.) through application of Copper hydroxide 

KEYWORD: ARRANGE IT ALPHABETICALLY

Introduction part is too lengthy try to reduce it

RESULT AND DISCUSSION: APPROPRIATE

CONCLUSION NEED TO REFRAMED AND ENHANCE THE CONCLUSION 

FIGURE- APPROPRIATE 

REFERENCES : FOLLOW THE JOURNAL GUIDELINES

Author Response

Dear,

Reviewer

Plants Journal

We appreciate the opportunity to revise and improve the manuscript following the reviewers’ suggestions. We are grateful for the opportunity to resubmit the manuscript entitled "Modification of sugar profile and ripening in atemoya (Annona × atemoya Mabb.) fruits through copper hydroxide application", n° Submission plants-2156385.

In this new letter we have tried to include all your suggestions and corrections, in order to improve the quality of our study. All reviewers’ suggestions were accepted, and all changes made are highlighted in yellow.

Below you will find the reviewer comments with our responses for each point written in blue color.

Sincerely yours.       

Caroline P. Cardoso, Felipe G. Campos, Gabriel M. Napoleão, Gustavo R. Barzotto, Lauro P. Campos, Gisela Ferreira and Carmen S. F. Boaro.

Reviewer 2

Title : need to reframed Modification of sugar profile and shelf life in atemoya (Annona × atemoya Mabb.) through application of Copper hydroxide

As suggested, we reframed the title. The term ripening from a physiological point of view is more appropriate.

KEYWORD: ARRANGE IT ALPHABETICALLY

As suggested, we arranged all keywords alphabetically.

Introduction part is too lengthy try to reduce it

As suggested, we reframed the introduction. Modified lines 102-104, as well as the abstract lines 18-27.

CONCLUSION NEED TO REFRAMED AND ENHANCE THE CONCLUSION

As suggested, we reframed the conclusion and enhanced it, lines 401-404.

In addition, some parts were ordered in different manner to a better understanding. Lines 236-237, 262-273, 277-278, 286-287, lines 333-339 referring all treatments in a table.

Reviewer 3 Report

I have the following questions and suggestions:

1.In abstract,line 26,"changed fruits acidity and decreased titratable acidity"is repeated.

2.In the introduction part, there are some contents that are not necessary, such as the relationship between sugar and disease resistance, acid, electron transport chain, ROS and so on.The detailed introduction of these contents leads to the lack of emphasis in the introduction section.

3.The treatment of T1-T5 will be shown more clearly in the way of graph or table.

4.line 156,DAFA. The full name should be clearly written when the acronym appears for the first time.

5.In figure 1, the ordinate name should be written directly to MDA.

6.What do the uppercase and lowercase letters in figure 3 mean?

7.Line 327-329, this part is inexplicable.

8.contributing to sweetness?In Fig3,there is no change in soluble solids.

9. In addition, there are many unprofessional aspects in the English expression of this article,such as,"plant's";line 343,Figure 9; etc. Please check and revise your article carefully.

Author Response

Dear,

Reviewer

Plants Journal

We appreciate the opportunity to revise and improve the manuscript following the reviewers’ suggestions. We are grateful for the opportunity to resubmit the manuscript entitled "Modification of sugar profile and ripening in atemoya (Annona × atemoya Mabb.) fruits through copper hydroxide application", n° Submission plants-2156385.

In this new letter we have tried to include all your suggestions and corrections, in order to improve the quality of our study. All reviewers’ suggestions were accepted, and all changes made are highlighted in yellow.

Below you will find the reviewer comments with our responses for each point written in blue color.

Sincerely yours.       

Caroline P. Cardoso, Felipe G. Campos, Gabriel M. Napoleão, Gustavo R. Barzotto, Lauro P. Campos, Gisela Ferreira and Carmen S. F. Boaro.

Reviewer 3

1.In abstract,line 26,"changed fruits acidity and decreased titratable acidity"is repeated.

As suggested, we reframed the abstract lines 18-27.

2.In the introduction part, there are some contents that are not necessary, such as the relationship between sugar and disease resistance, acid, electron transport chain, ROS and so on. The detailed introduction of these contents leads to the lack of emphasis in the introduction section.

As suggested, we deleted some extra parts that are not necessary.

3.The treatment of T1-T5 will be shown more clearly in the way of graph or table.

As suggested, we reframed how we presented the treatments in a table (lines 333-339).

4.line 156,DAFA. The full name should be clearly written when the acronym appears for the first time.

As suggested, it’s been modified.

5.In figure 1, the ordinate name should be written directly to MDA.

As suggested, it’s been modified.

6.What do the uppercase and lowercase letters in figure 3 mean?

Upper and lowercase letters are discriminated now, 286-287.

7.Line 327-329, this part is inexplicable.

As suggested, we reframed the sentence, lines 277-278.

8.contributing to sweetness?In Fig3,there is no change in soluble solids.

All dissolved soluble solids resulted in a determined refraction (ºBrix). Determined sugars revealed high concentration of sucrose and trehalose, that can be related to fruits sweetness. However, in the present study, the sugar profile was determined by high performance ion chromatography, which may explain the lack of correlation with the soluble solids, determined for the calculation of the maturation index. This index, which revealed an inverse correlation with the sugar profile, probably due to the titratable acidity, delayed fruit ripening and increased shelf life. Lines 236-237, 262-273.

  1. In addition, there are many unprofessional aspects in the English expression of this article,such as,"plant's";line 343,Figure 9; etc. Please check and revise your article carefully.

We sent the article to a English revisor after all suggestions have been made and the certificate is attached.

In addition, some parts were ordered in different manner to a better understanding. Lines 102-104 and lines 401-404 reframing the conclusion.

Round 2

Reviewer 1 Report

Thank you for your hard work in revising the paper you submitted by referring to the contents of the 1st review.

It was confirmed that the contents of the review were reflected and corrected.

However, it is judged that a considerable number of references will not be cited as some contents have been deleted from the introduction, results, and discussion.

Please check again the references cited in the manuscript.

Reviewer 3 Report

The author basically solved all my problems.Thank you.